# VisionReasoner: Unified Reasoning-Integrated Visual Perception via Reinforcement Learning

**Yuqi Liu**[1*] **Tianyuan Qu**[1*] **Zhisheng Zhong**[1] **Bohao Peng**[1] **Shu Liu**[2✉] **Bei Yu**[1] **Jiaya Jia**[2,3]

The Chinese University of Hong Kong[1]    SmartMore[2]

The Hong Kong University of Science and Technology[3]    *Equal ✉Correspondence

## Abstract

Large vision-language models exhibit inherent capabilities to handle diverse visual perception tasks. In this paper, we introduce VisionReasoner, a unified framework capable of reasoning and solving multiple visual perception tasks within a shared model. Specifically, by designing a unified reward mechanism and multi-object cognitive learning strategies, VisionReasoner enhances its reasoning capabilities to analyze visual inputs, and addresses diverse perception tasks within a unified model. VisionReasoner generates a structured reasoning process before delivering the desired outputs responding to user queries. Human evaluation reveals the reasoning process of VisionReasoner is faithful and reliable even without annotated reasoning train data. To rigorously assess unified visual perception capabilities, we evaluate VisionReasoner on ten diverse tasks spanning three critical domains: detection, segmentation, and counting. Experimental results show that Vision-Reasoner achieves superior performance as a unified model, outperforming the baseline Qwen2.5VL by relative margins of 29.1% on COCO (detection), 22.1% on ReasonSeg (segmentation), and 13.2% on CountBench (counting). Code 🎁

## 1 Introduction

Recent advances in large vision-language models (LVLMs) (Bai et al., 2025; Wang et al., 2024; Google, 2025; OpenAI, 2025) have demonstrated remarkable capabilities in visual conversations. As the field progresses, researchers are increasingly applying LVLMs to a wider range of visual perception tasks, such as visual grounding (Peng et al., 2024) and reasoning segmentation (Lai et al., 2024; Liu et al., 2025a) , often incorporating task-specific modules or techniques.

Through an analysis of diverse visual perception tasks, we observe that many can be categorized into three fundamental types: detection (e.g., object detection (Lin et al., 2014), visual grounding (Yu et al., 2016)), segmentation (e.g., referring expression segmentation (Yu et al., 2016), reasoning segmentation (Lai et al., 2024)), and counting (e.g., object counting (Paiss et al., 2023)). Notably, our analysis reveals that these three task types share a common structure as multi-object cognition problems, suggesting that they can be addressed through a unified framework.

Moreover, recent studies have explored the integration of reinforcement learning (RL) into LVLMs (Team, 2025; Liu et al., 2025b;a; Zheng et al., 2025). Works such as VisualRFT (Liu et al., 2025b) and Seg-Zero (Liu et al., 2025a) demonstrate that RL can enhance reasoning in visual perception tasks. However, these approaches often employ RL in a task-specific manner, training with different data for different tasks, which may limit their scalability and generalizability.

Building on these insights, we propose VisionReasoner, a unified framework that addresses diverse visual perception tasks through a shared architecture. The framework's core capabilities, which include advanced reasoning and multi-object cognition, are enabled through RL and a unified reward mechanism. Format rewards, including thinking rewards that promote structured reasoning and non-repeat rewards that prevent redundant reasoning patterns. Accuracy rewards, comprising multi-object IoU rewards and L1 rewards for precise localization, strengthen multi-object cognition. Unlike previous approaches like Kosmos (Peng et al., 2024) that use cross-entropy loss, our RL framework requires optimal prediction-to-ground-truth matching. We address this challenge by implementing an efficient matching pipeline combining the batch computing and the Hungarian algorithm, significantly improving computational efficiency while maintaining matching accuracy.

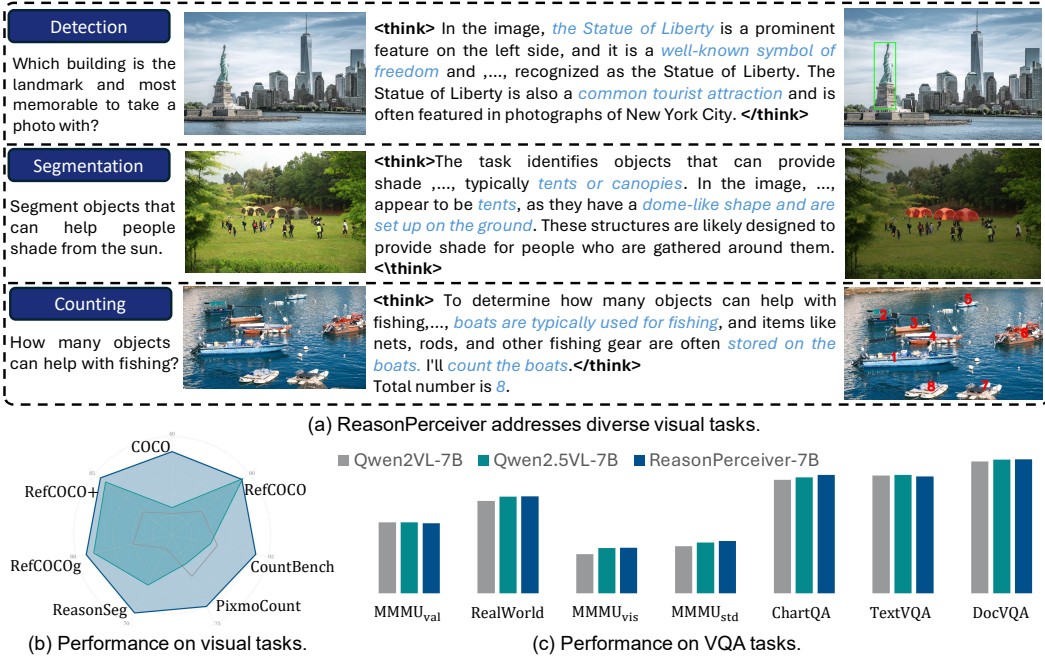

Figure 1: (a) VisionReasoner addresses diverse tasks within a unified framework. It generates a reasoning process and outputs the expected result corresponding to each query. (b) VisionReasoner significantly outperforms Qwen2.5VL. (c) VisionReasoner retains strong VQA capabilities.

To comprehensively evaluate model performance, we conduct extensive experiments with VisionReasoner across 10 diverse tasks spanning three fundamental types: detection, segmentation, and counting. Remarkably, our VisionReasoner-7B model achieves strong performance despite being trained on only 7k samples, demonstrating both robust test-time reasoning capabilities and effective multi-task generalization, as shown in Figure 1 (a)-(b). Experimental results show significant improvements over baseline models, with relative gains of 29.1% on COCO-val (detection), 22.1% on ReasonSeg-test (segmentation), and 13.2% on CountBench-test (counting), validating the effectiveness of our unified approach. Additionally, VisionReasoner exhibits visual question answering capabilities comparable to state-of-the-art models, as shown in Figure 1 (c). Human evaluation also indicates VisionReasoner generates faithful and reliable reasoning process even without training on annotated reasoning data.

Our contributions are summarized as follows:

- We propose VisionReasoner, a unified framework for visual perception tasks. Through carefully crafted rewards and training strategy, VisionReasoner has strong multi-task capability, addressing diverse visual perception tasks within a shared model.
- Experimental results show that VisionReasoner achieves superior performance across ten diverse visual perception tasks within a single unified framework, outperforming baseline models by a significant margin.
- Through extensive ablation studies, we validate the effectiveness of our design and offer critical insights into the application of RL in LVLMs.

## 2 RELATED WORKS

### 2.1 LARGE VISION-LANGUAGE MODELS

Following LLaVA's (Liu et al., 2023c) pioneering work on visual instruction tuning for large vision-language models, subsequent studies (Wang et al., 2024; Meta, 2024; OpenAI, 2025; Bai et al., 2025; Li et al., 2024b; Zhong et al., 2024) have adopted this paradigm for vision-language conversation. Beyond visual conversation tasks, LVLMs have been extended to diverse vision applications, including

visual grounding (Peng et al., 2024) and reasoning segmentation (Lai et al., 2024). Notely, the recent GPT-4.1 (OpenAI, 2025) demonstrates state-of-the-art performance in multi-modal information processing and visual reasoning. Although these models are evaluated on specific tasks, their performance has not been systematically evaluated under a unified visual perception framework.

## 2.2 REINFORCEMENT LEARNING IN LARGE MODELS

In the field of large language model (LLMs), various reinforcement learning (RL) algorithms are used to enhance model performance, such as reinforcment learning from human feedback (RLHF) (Ouyang et al., 2022), direct preference optimization (DPO) (Rafailov et al., 2023) and proximal policy optimization (PPO) (Schulman et al., 2017). The recent DeepSeek R1 (Guo et al., 2025), trained using Group Relative Policy Optimization (GRPO) (Shao et al., 2024), demonstrates remarkable test-time scaling capabilities, significantly improving reasoning ability and overall performance. Building on these advances, researchers try to apply these RL techniques to LVLMs. Notable efforts include Visual-RFT (Liu et al., 2025b), EasyR1 (Zheng et al., 2025) and Seg-Zero(Liu et al., 2025a), all of which exhibit strong reasoning capabilities and achieve impressive performance.

## 3 METHOD

To develop a unified visual perception model capable of solving diverse vision tasks, we identify and analyze the representative visual perception tasks, then reformulate their inputs and outputs into a set of three fundamental task categories (Section 3.2). Next, we detail the architecture of our VisionReasoner model (Section 3.3). Additionally, we present the unified reward mechanism employed for training our model (Section 3.4). Finally, we elaborate on our training strategy of multi-object cognition (Section 3.5).

## 3.1 PRELIMINARY

**Traditional Vision Methods vs. LVLMs.** Although traditional vision models (Cheng et al., 2024; Ren et al., 2024a) achieve strong performance on standard visual perception benchmarks (Lin et al., 2014), they are inherently limited to processing simple categorical queries and struggle with complex, compositional, or reasoning-intensive instructions. In contrast, LVLMs can interpret and respond to nuanced, open-ended queries. As illustrated in Figure 2, VisionReasoner successfully localizes and identifies target objects that traditional approaches fail to detect, highlighting the necessity to integrate LVLMs into visual perception pipelines where reasoning are essential.

**Group Relative Policy Optimization (GRPO).** The GRPO is a on-policy reinforcement learning algorithm. For each input $x$, the old policy model $\pi_{\theta_{old}}$ from previous step generate a group of rollouts $\{o_i\}_{i=1}^{G}$. Then reward functions are used to calculate rewards for each $o_i$, getting $\{r_i\}_{i=1}^{G}$. We design a unified reward mechanism and the relative advantage is calculated as:

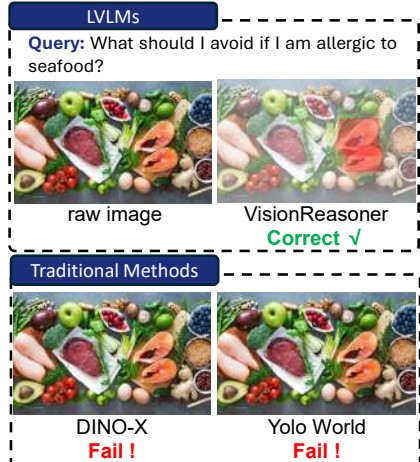

Figure 2: VisionReasoner correctly localizes objects from a complex instruction, whereas both commercial DINO-X and open-source YOLO-World fail.

$$A_i = \frac{r_i - \text{mean}(\{r_1, r_2, \ldots, r_G\})}{\text{std}(\{r_1, r_2, \ldots, r_G\})}. \quad (1)$$

The GRPO maximizes the following objective and optimizes the model $\pi_\theta$:

$$\mathcal{J}_{\text{GRPO}}(\theta) = \mathbb{E}_{x \sim \text{Train Batch}, \{o_i\}_{i=1}^{G} \sim \pi_{\theta_{\text{old}}}(O|x)}$$

$$\left[ \frac{1}{G} \sum_{i=1}^{G} \min\left( \frac{\pi_\theta(o_i \mid x)}{\pi_{\theta_{\text{old}}}(o_i \mid x)} A_i, \text{clip}\left( \frac{\pi_\theta(o_i \mid x)}{\pi_{\theta_{\text{old}}}(o_i \mid x)}, 1-\varepsilon, 1+\varepsilon \right) A_i \right) - \beta D_{\text{KL}}(\pi_\theta \parallel \pi_{\text{ref}}) \right]. \quad (2)$$

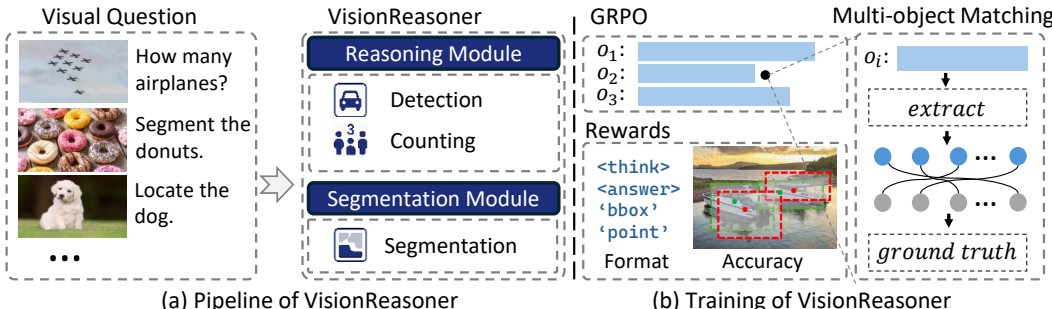

Figure 3: Illustration of VisionReasoner. (a) For a given image $\mathbf{I}$ and text instruction $\mathbf{T}$, our model generates the expected output corresponding to the instruction. (b) For each observation $o_i$, we calculate the rewards (Section 3.4) and attain the optimal match of multi-objects (Section 3.5)

## 3.2 TASK REFORMULATION AND CATEGORIZATION

Our analysis of vision perception tasks (Yu et al., 2016; Lin et al., 2014; Lai et al., 2024; Deitke et al., 2024) reveals that many of them can be categorized into three fundamental task types. Here we take ten visual perception tasks for illustration. Further details are provided in the Appendix J.

**Detection.** Given an image $\mathbf{I}$ and a text query $\mathbf{T}$, the detection task type aims to generate a set of bounding boxes $\{\mathbf{B}_i\}_{i=1}^N$ that localize objects of interest. This type requires multi-object cognition ability. This category includes tasks such as Visual Grounding (Yu et al., 2016; Kazemzadeh et al., 2014) and Object Detection (Lin et al., 2014).

**Segmentation.** Given an image $\mathbf{I}$ and a text query $\mathbf{T}$, the segmentation task type aims to generate a set of binary segmentation masks $\{\mathbf{M}_i\}_{i=1}^N$ that identify the regions of interest. We address this type by detect-then-segment paradigm. This category includes tasks such as Referring Expression Segmentation (Kazemzadeh et al., 2014; Yu et al., 2016) and Reasoning Segmentation (Lai et al., 2024; Yang et al., 2023).

**Counting.** Given an image $\mathbf{I}$ and a text query $\mathbf{T}$, the counting task type aims to estimate the number of target objects specified by the query. We address this type by detect-then-count paradigm. This category includes tasks such as Object Counting (Deitke et al., 2024; Paiss et al., 2023).

## 3.3 VISIONREASONER MODEL

Our VisionReasoner $\mathcal{F}$ model incorporates a reasoning module, which processing image and locates targeted objects, and a segmentation module that produces segmentation masks if needed. The whole architecture is shown in Figure 3 (a). The key to $\mathcal{F}$ lies in its multi-object cognition capabilities, which is critical to enables VisionReasoner to address three fundamental task types: detection, segmentation, counting. Specifically, given an image $\mathbf{I}$, a text query $\mathbf{T}$, the VisionReasoner $\mathcal{F}$ generates an interpretable reasoning process, and then produces the bounding boxes $\{\mathbf{B}_i\}_{i=1}^N$ and central points $\{\mathbf{P}_i\}_{i=1}^N$ of targeted objects corresponding to $\mathbf{T}$. Then $\{\mathbf{B}_i\}_{i=1}^N$ and $\{\mathbf{P}_i\}_{i=1}^N$ serve as bridge to connect the segmentation module, producing binary masks $\{\mathbf{M}_i\}_{i=1}^N$ if needed. This process can be formulated as:

$$(\{\mathbf{B}_i, \mathbf{M}_i\})_{i=1}^N = \mathcal{F}(\mathbf{I}, \mathbf{T}).\tag{3}$$

During inference, the user provide the input image $\mathbf{I}$ and text prompt $\mathbf{T}$, and define a specified task type $\mathbf{C} \in \{\text{detection}, \text{segmentation}, \text{counting}\}$. The system then produces the expected outputs as follows:

$$\text{Output} = \begin{cases} \{\mathbf{B}_i\}_{i=1}^N, & \text{if } \mathbf{C} \text{ is detection}, \\ \{\mathbf{M}_i\}_{i=1}^N, & \text{if } \mathbf{C} \text{ is segmentation}, \\ N, & \text{if } \mathbf{C} \text{ is counting}. \end{cases}\tag{4}$$

In this way, our VisionReasoner can process diverse perception tasks in a unified manner within a shared framework. Moreover, our framework can be easily extended to other visual perception tasks as illustrated in Appendix H.

## 3.4 UNIFIED REWARD MECHANISM

As illustrated in Section 3.1, the core in group relative RL is the design of rewards. We design a unified reward mechanism for visual perception tasks, including format rewards and accuracy rewards. We use target object bboxes and center points to calculate the rewards rather than binary masks. These rewards jointly guide the optimization process by reinforcing both structural correctness and multi-object recognition performance. The model is capable of addressing diverse visual perception tasks after training within this unified reward mechanism. The total reward is the sum of all rewards.

**Thinking Format Reward.** This reward is 1.0 if the model output a thinking process between <think> and </think>tags, and output the final answer between the <answer> and </answer>tags.

**Answer Format Reward.** We use bounding boxes $\{\mathbf{B}_i\}_{i=1}^N$ and points $\{\mathbf{P}_i\}_{i=1}^N$ as the answer as it has better training efficiency. So this reward restrict the model output answer in $[\{'\text{bbox\_2d}' : [x_1, y_1, x_2, y_2],' \text{point\_2d}' : [x_1, y_1]\}, ...]$. The reward is 1.0 if correct else 0.0.

**Non Repeat Format Reward.** We split the reasoning process into sentences to detect repeated pattern. A reward of 1.0 is assigned for those with unique or non-repetitive thinking processes.

**Bboxes IoU Reward.** Given a set of $N$ ground-truth bounding boxes and $K$ predicted bounding boxes, this reward computes their optimal one-to-one matched Intersection-over-Union (IoU) scores. For each IoU exceeding 0.5, we increment the reward by $\dfrac{1}{\max\{N, K\}}$.

**Bboxes L1 Reward.** Given a set of $N$ ground-truth bounding boxes and $K$ predicted bounding boxes, this reward computes their one-to-one matched L1 distance. For each L1 distance below the threshold of 10 pixel, we increment the reward by $\dfrac{1}{\max\{N, K\}}$.

**Points L1 Reward.** Given a set of $N$ ground-truth points and $K$ predicted points, this reward computes their one-to-one matched L1 distance. For each L1 distance below the threshold of 30 pixel, we increment the reward by $\dfrac{1}{\max\{N, K\}}$.

## 3.5 MULTI-OBJECT COGNITION IN LVLMS

Unlike the auto-regressive training paradigm (Peng et al., 2024; Bai et al., 2025) in supervised fine-tuning, RL framework requires optimal prediction-to-ground-truth matching for reward calculation. To address this, we derive the necessary data and implement an effective matching strategy.

**Multi-object Data Preparation.** We derive bboxes and points directly from the original mask annotations in existing segmentation datasets (e.g., RefCOCOg (Yu et al., 2016), LISA++(Yang et al., 2023)). Specifically, for a given binary mask of an object, we determine its bounding box by extracting the leftmost, topmost, rightmost, and bottommost pixel coordinates. Additionally, we compute the center point coordinates of the mask. We process multiple objects per image by: (i) using one central point (ii) joining all textual descriptions with the conjunction 'and', and (iii) concatenating all associated bounding boxes and center points into list per image.

---

**Algorithm 1:** Multi-object Matching

**Input:** pred bboxes $\boldsymbol{b}_{\text{pred}} \in \mathbb{R}^{K \times 4}$; pred points $\boldsymbol{p}_{\text{pred}} \in \mathbb{R}^{K \times 2}$;
       GT bboxes $\boldsymbol{b}_{\text{gt}} \in \mathbb{R}^{N \times 4}$; GT points $\boldsymbol{p}_{\text{gt}} \in \mathbb{R}^{N \times 2}$

**Function** *AccuracyReward($\boldsymbol{b}_{pred}, \boldsymbol{p}_{pred}, \boldsymbol{b}_{gt}, \boldsymbol{p}_{gt}$):*
  $r \leftarrow 0; L_{\max} \leftarrow \max(K, N);$
  $IoU \leftarrow \text{BatchIoU}(\boldsymbol{b}_{\text{pred}}, \boldsymbol{b}_{\text{gt}}) \in \mathbb{R}^{K \times N}$
  $BL1 \leftarrow \text{BatchBoxL1Distance}(\boldsymbol{b}_{\text{pred}}, \boldsymbol{b}_{\text{gt}}) \in \mathbb{R}^{K \times N}$
  $PL1 \leftarrow \text{BatchPointL1Distance}(\boldsymbol{p}_{\text{pred}}, \boldsymbol{p}_{\text{gt}}) \in \mathbb{R}^{K \times N}$
  $R_{\text{IoU}} \leftarrow [IoU > \text{IoU threshold}]$
  $R_{\text{BL1}} \leftarrow [BL1 < \text{Box L1 threshold}]$
  $R_{\text{PL1}} \leftarrow [PL1 < \text{Point L1 threshold}]$
  $C \leftarrow (3 - (R_{\text{IoU}} + R_{\text{BL1}} + R_{\text{PL1}})) \in \mathbb{R}^{K \times N}$
  $(\boldsymbol{r}, \boldsymbol{c}) \leftarrow \texttt{Hungarian}(C)$
  $\text{total} \leftarrow 3|\boldsymbol{r}| - \sum_t C_{\boldsymbol{r}_t, \boldsymbol{c}_t}$
  $r \leftarrow \text{total}/L_{\max};$ **return** $r$

**Output:** Accuracy reward $r$

---

**Multi-object Matching.** Our framework addresses multi-object matching through batch computation and the Hungarian algorithm, which optimally solves the many-to-many matching problem for bounding boxes IoU rewards, bounding boxes L1 rewards, and points L1 rewards. As shown in

Table 1: Performance comparison on detection tasks.

| Method | Detection | | | | | | | Avg. |
|---|---|---|---|---|---|---|---|---|
| | COCO | RefCOCO | | RefCOCO+ | | RefCOCOg | | |
| | val | val | testA | val | testA | val | test | |
| *Task-specific Models* | | | | | | | | |
| VGTR | - | 79.0 | 82.3 | 63.9 | 70.1 | 65.7 | 67.2 | - |
| TransVG | - | 81.0 | 82.7 | 64.8 | 70.7 | 68.7 | 67.7 | - |
| RefTR | - | 85.7 | 88.7 | 77.6 | 82.3 | 79.3 | 80.0 | - |
| MDETR | - | 86.8 | 89.6 | 79.5 | 84.1 | 81.6 | 80.9 | - |
| OWL-ViT | 30.9 | - | - | - | - | - | - | - |
| YOLO-World-S | 37.6 | - | - | - | - | - | - | - |
| GLIP-T | 46.6 | 50.4 | 54.3 | 49.5 | 52.8 | 66.1 | 66.9 | 55.2 |
| G-DINO-T | 48.4 | 74.0 | 74.9 | 66.8 | 69.9 | 71.1 | 72.1 | 68.2 |
| DQ-DETR | 50.2 | 88.6 | 91.0 | 81.7 | 86.2 | 82.8 | 83.4 | **80.6** |
| *Large Vision-language Models* | | | | | | | | |
| Shikra-7B | - | 87.0 | 90.6 | 81.6 | 87.4 | 82.3 | 82.2 | - |
| InternVL2-8B | - | 87.1 | 91.1 | 79.8 | 87.9 | 82.7 | 82.7 | - |
| Qwen2-VL-7B | 28.3 | 80.8 | 83.9 | 72.5 | 76.5 | 77.3 | 78.2 | 71.1 |
| Qwen2.5-VL-7B | 29.2 | 88.8 | 91.7 | 82.3 | 88.2 | 84.7 | 85.7 | 78.6 |
| VisionReasoner-7B | 37.7 | 88.6 | 90.6 | 83.6 | 87.9 | 86.1 | 87.5 | **80.3** |

Figure 3 (b), for each observation $o_j$, which contains a list of bboxes $\{\mathbf{B}_{\mathbf{pred}_i}\}_{i=1}^{K}$ and points $\{\mathbf{P}_{\mathbf{pred}_i}\}_{i=1}^{K}$, we calculate its reward scores with the ground-truth bboxes $\{\mathbf{B}_{\mathbf{GT}i}\}_{i=1}^{N}$ and points $\{\mathbf{P}_{\mathbf{GT}i}\}_{i=1}^{N}$ by implementing batch computation. We then calculate the optimal one-to-one matching with using Hungarian algorithm. The pseudocode of multi-object matching is shown in Algorithm 1. These design guarantees optimal assignment between predictions and ground truth annotations while achieving high computational efficiency.

# 4 EXPERIMENTS

## 4.1 EXPERIMENTAL SETTINGS

**Evaluation Benchmark.** We use ten benchmarks to evaluate model performance across general vision perception tasks, including three fundamental task types: detection, segmentation and counting. Specifially, we employ COCO (Lin et al., 2014) and RefCOCO(+/g) (Yu et al., 2016) for detection evaluation; RefCOCO(+/g) and ReasonSeg (Lai et al., 2024) for segmentation evaluation; PixMo-Count (Deitke et al., 2024) and CountBench (Paiss et al., 2023) for counting evaluation. Details of benchmarks and metrics can be found in Appendix B.

**Training Data.** The training data is sourced from the training splits of four datasets: LVIS (Gupta et al., 2019), RefCOCOg (Yu et al., 2016), gRefCOCO (Liu et al., 2023a), and LISA++ (Yang et al., 2023). We randomly collect approximately 7k training samples, with around 1,800 from each dataset.

**Implementation Details.** We initialize VisionReasoner with Qwen2.5-VL and SAM2. We employ a batch size of 16 and a learning rate of 1e-6. The training objective is Equation (2).

## 4.2 MAIN RESULTS

We compare the results with LVLMs and task-specific models on each of the three fundamental task types. It is worthy note that our VisionReasoner is capable of handling different tasks within the same model and is evaluated in a zero-shot manner.

**Detection.** We compare VisionReasoner with several state-of-the-art LVLMs, including Shikra (Chen et al., 2023), InternVL2-8B (OpenGVLab, 2024), Qwen2-VL-7B (Wang et al., 2024) and Qwen2.5VL-7B (Bai et al., 2025). For task-specific models, we evaluate against VGTR (Da et al., 2023), TransVG (Deng et al., 2021), RefTR (Li & Sigal, 2021), MDETR (Kamath et al., 2021), OWL-ViT (Minderer et al., 2022), YOLO-World (Cheng et al., 2024), GroundingDINO (Liu et al., 2024a), DQ-DETR (Liu

Table 2: Performance comparison on segmentation tasks and counting tasks. We use SAM2 for vision-language models if necessary in segmentation tasks.

| Method | Segmentation | | | | Avg. | Counting | | | Avg. |
|---|---|---|---|---|---|---|---|---|---|
| | ReasonSeg | RCO | RCO+ | RCOg | | Pixmo | | Count | |
| | val | test | testA | testA | test | | val | test | test | |
| *Task-specific Models* | | | | | | | | | | |
| LAVT | - | - | 75.8 | 68.4 | 62.1 | - | - | - | - | - |
| ReLA | 22.4 | 21.3 | 76.5 | 71.0 | 66.0 | 51.4 | - | - | - | - |
| *Large Vision-language Models* | | | | | | | | | | |
| LISA-7B | 44.4 | 36.8 | 76.5 | 67.4 | 68.5 | 58.7 | - | - | - | - |
| LLaVA-OV-7B | - | - | - | - | - | - | 55.8 | 53.7 | 78.8 | 62.8 |
| GLaMM-7B | - | - | 58.1 | 47.1 | 55.6 | - | - | - | - | |
| PixelLM-7B | - | - | 76.5 | 71.7 | 70.5 | - | - | - | - | |
| Seg-Zero-7B | 62.6 | 57.5 | 80.3 | 76.2 | 72.6 | 69.8 | - | - | - | |
| Qwen2-VL-7B | 44.5 | 38.7 | 68.7 | 65.7 | 63.5 | 56.2 | 61.6 | 56.3 | 80.4 | 66.1 |
| Qwen2.5-VL-7B | 56.9 | 52.1 | 79.9 | 76.8 | 72.8 | 67.7 | 58.1 | 53.1 | 78.8 | 63.6 |
| VisionReasoner-7B | 66.3 | 63.6 | 78.9 | 74.9 | 71.3 | **71.0** | 70.1 | 70.7 | 89.2 | **76.7** |

Table 3: Comparison of multi-object matching. Our code achieves a $4\times$ speedup.

| Hungarian | BatchComp | Time (s) |
|---|---|---|
| ✓ | | $2 \times 10^{-3}$ |
| ✓ | ✓ | $5 \times 10^{-4}$ |

Table 4: Comparison on the reasoning length.

| Data | Avg. Len (# words) |
|---|---|
| COCO | 62 |
| RefCOCOg | 65 |
| ReasonSeg | 71 |

Table 5: Comparison on different RL algorithm.

| RL | ReasonSeg-val |
|---|---|
| Baseline | 56.9 |
| GRPO | 61.9 |
| DAPO | 61.7 |

et al., 2023d), GLIP (Li et al., 2022). Since LVLMs do not output confidence score, we approximate it using the ratio of the bounding box area to the total image area (bbox_area / image_area) to enable compatibility with COCOAPI (Team, 2014). However, this coarse approximation leads to underestimated AP scores. As shown in Table 1, VisionReasoner achieves superior performance among LVLMs. While our model shows a performance gap compared to some task-specific baselines on COCO datasets, it maintains competitive advantages due to its superior generalization capability.

**Segmentation.** We evaluate VisionReasoner against state-of-the-art LVLMs, including LISA (Lai et al., 2024), GLaMM (Rasheed et al., 2024), PixelLM (Ren et al., 2024b), Seg-Zero (Liu et al., 2025a), Qwen2-VL (Wang et al., 2024) and Qwen2.5VL (Bai et al., 2025). For these LVLMs, we first extract bounding box predictions and subsequently send them into SAM2(Ravi et al., 2024) to generate segmentation masks. We also compare task-specific models, including LAVT (Yang et al., 2022) and ReLA (Liu et al., 2023b). For models that do not report gIoU, we report their cIoU as an alternative. As shown in Table 2, VisionReasoner achieves state-of-the-art performance, outperforming both general-purpose LVLMs and task-specific approaches.

**Counting.** We evaluate VisionReasoner against state-of-the-art LVLMs, including LLaVA-OneVision (Li et al., 2024a), Qwen2-VL-7B (Wang et al., 2024) and Qwen2.5VL-7B (Bai et al., 2025). We evaluate these LVLMs in a first-detect-then-count manner. As shown in Table 2, VisionReasoner achieves state-of-the-art performance.

### 4.3 ABLATION STUDY

We perform ablation studies to assess the effectiveness of our design and validate the optimal hyper-parameter selection and training recipe design for VisionReasoner. We also evaluate VisionReasoner on VQA tasks.

**Multi-object Matching.** We quantitatively assess the efficiency of our two key design choices for multi-object matching: the Hungarian algorithm and batch computation. In a scenario with 30 objects, Table 3 demonstrates that a non-batch matching require $2 \times 10^{-3}$ seconds to complete, while our optimized approach achieves matching in just $5 \times 10^{-4}$ seconds - a $4\times$ speedup.

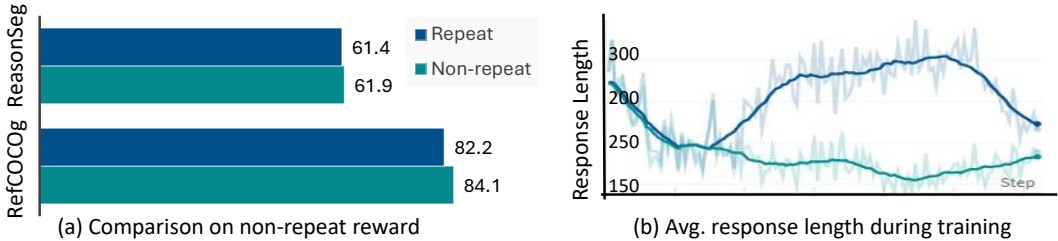

Figure 4: Ablation on non-repeat reward. (a) Consistent performance gain across different datasets using non-repeated reward. (b) Non-repeat rewards lead to shorter response lengths.

Table 6: Performance comparison on different training data.

| Method | Training Data | | | | Det | Seg | Avg. |
|---|---|---|---|---|---|---|---|
| | RefCOCOg | gRefCOCO | LVIS | LISA++ | RefCOCOg-val | ReasonSeg-val | |
| VisionReasoner-7B | ✓ | | | | 84.1 | 61.9 | 73.0 |
| | ✓ | ✓ | | | 85.8 | 63.8 | 74.8 |
| | ✓ | ✓ | ✓ | | 85.5 | 64.2 | 74.9 |
| | ✓ | ✓ | ✓ | ✓ | **86.1** | **66.3** | **76.2** |

Table 7: Performance comparison on VQA tasks.

| Method | OCRBench | RealworldQA | MMMUPro$_{vision}$ | MMMUPro$_{std}$ | ChartQA | DocVQA |
|---|---|---|---|---|---|---|
| Qwen2VL-7B | 809 | 66.1 | 28.0 | 33.8 | 81.4 | 94.5 |
| Qwen2.5VL-7B | 822 | 69.2 | 32.4 | 36.4 | 83.1 | 95.7 |
| VisionReasoner-7B | **825** | **69.5** | **32.6** | **37.4** | **84.9** | **96.0** |

**Reasoning Length.** As shown in Table 4, our analysis reveals that the model's reasoning length adapts dynamically to text query complexity. Specifically, for simple class names in COCO and short phrases in RefCOCOg, the reasoning process is relatively concise. In contrast, complex reasoning-intensive queries in ReasonSeg require longer reasoning processes.

**Non Repeat Reward.** Figure 4 (a) presents the performance comparison with and without the non-repeat reward. Models are trained only on 2,000 samples from RefCOCOg. The model achieves better results when trained using the non-repeat reward. Additionally, model without non-repeat reward tends to generate longer reasoning processes, as shown in Figure 4 (b), and we observe repetitive reasoning patterns during inference.

**Different RL Algorithm.** We use different on-policy RL training algorithm: the GRPO (Shao et al., 2024) and DAPO (Yu et al., 2025). Models are trained only on 2,000 samples from RefCOCOg. As shown in Table 5, performance consistently improves across both algorithms, demonstrating that our training framework is both stable and generalizable.

**Different Training Data.** We conduct an ablation study on different training datasets, with results presented in Table 6. The four datasets provide diverse text annotations: LVIS uses simple class names, RefCOCOg contains single-object referring expressions, gRefCOCO includes expressions that may refer to multiple objects, and LISA++ features texts requiring reasoning. Our experiments demonstrate that these datasets consistently improve model performance.

**Visual QA Ability.** We also compare VisionReasoner's VQA (Masry et al., 2022; Mathew et al., 2021; Liu et al., 2024b; Yue et al., 2024; xAI, 2024) ability with Qwen2VL (Wang et al., 2024) and the baseline model Qwen2.5VL(Bai et al., 2025). As shown in Table 7, VisionReasoner achieves a slight performance gain even though we do not train on VQA data.

**Sampling Number.** Figure 5 presents performance comparison with different sampling number. Models are trained using all 7k training samples. We observe an initial performance gain followed by a notable decline with larger sampling number, suggesting that excessive sampling may induce overfitting to the training distribution and consequently degrade generalization capability.

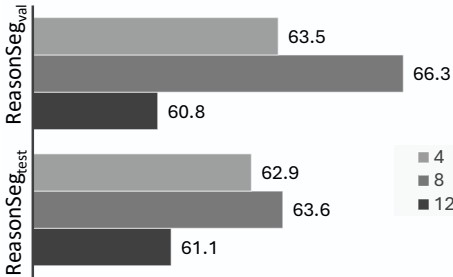

Figure 5: Different sampling number.

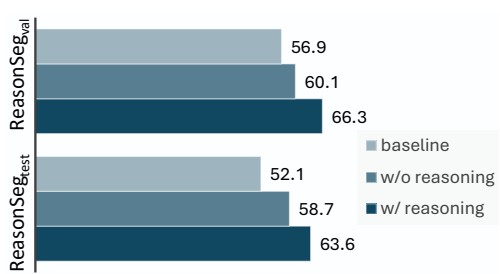

Figure 6: Reasoning vs. no reasoning

**Reasoning.** Figure 6 compares the performance of models with and without reasoning. We eliminate reasoning process by removing thinking instruction and thinking reward. Models are trained using all 7k training samples. Our results show that both approaches outperform the baseline. And the reasoning-enhanced model demonstrates significant gain on intricate reasoning segmentation data.

## 4.4 HUMAN EVALUATION ON REASONING PROCESS

We employ three experts to conduct human evaluations on ReasonSeg-val to assess both answer consistency and image consistency of the reasoning trace. Image consistency measures whether the reasoning trace accurately describes the visual content in the image. Answer consistency evaluates whether the objects and information mentioned in the reasoning trace are consistent with the final predicted output. These evaluations help assess the faithfulness and reliability of the model's internal reasoning.

We evaluate the reasoning traces at different levels based on IoU (between prediction and ground-truth) values and results are shown in Table 8. We find that the overall image consistency accuracy and answer consistency accuracy reach 97.0% and 90.5%, respectively. The majority of problematic reasoning traces are concentrated in cases where the IoU is below 0.25. These results demonstrate that the reasoning trace of VisionReasoner is accurate and well-grounded, even though the model was trained without human-annotated reasoning data.

Table 8: Reasoning Process Analysis by IoU Range. IC: Image Consistency; AC: Answer Consistency.

| IoU_Range | Num | IC (%) | AC (%) |
|---|---|---|---|
| 0–0.25 | 26 | 76.9 | 46.2 |
| 0.25–0.50 | 29 | 100.0 | 93.1 |
| 0.50–0.75 | 35 | 100.0 | 91.4 |
| 0.75–1.00 | 110 | 100.0 | 100.0 |
| **ALL** | **200** | **97.0** | **90.5** |

## 4.5 QUALITATIVE RESULTS

We visualize some results on Figure 7. Notably, VisionReasoner addresses several visual perception tasks within a shared model. More results and the reasoning process are provided in Appendix G.

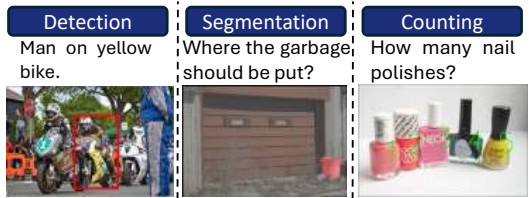

Figure 7: Qualitative results on different tasks. Reasoning process and more results are provided in Appendix G.

## 5 CONCLUSION

We present VisionReasoner, a unified vision-language framework for reasoning visual perception tasks. By introducing novel multi-object cognitive learning strategies and curated reward functions, VisionReasoner demonstrates strong capabilities in analyzing visual inputs, generating structured reasoning processes and delivering task-specific outputs. Experiments across ten diverse tasks, spanning detection, segmentation and counting, validates the robustness and versatility of our approach. Notably, VisionReasoner achieves significant improvements over baseline, with relative performance gains of 29.1% on COCO (detection), 22.1% on ReasonSeg (segmentation), and 13.2% on CountBench (counting). Human evaluation further reveals the reasoning traces of VisionReasoner are well-grounded and faithful.

## ACKNOWLEDGEMENTS

This work was supported in part by the Research Grants Council under the Areas of Excellence scheme grant AoE/E-601/22-R.

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

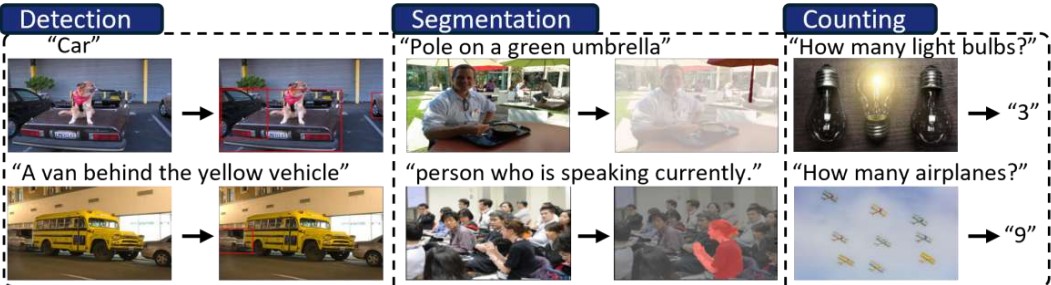

Figure 8: Examples from evaluation benchmarks. Zoom in for better viewing.

# A  THE USE OF LARGE LANGUAGE MODELS (LLMs)

LLMs are used only to polish writing in this paper.

# B  DETAILS OF EVALUATION BENCHMARKS

We use ten benchmarks to evaluate model performance across general vision perception tasks. Our evaluation includes three fundamental task types: detection, segmentation and counting. Specifically, we employ COCO (Lin et al., 2014) and RefCOCO(+/g) (Yu et al., 2016) for detection evaluation; RefCOCO(+/g) and ReasonSeg (Lai et al., 2024) for segmentation evaluation; PixMo-Count (Deitke et al., 2024) and CountBench (Paiss et al., 2023) for counting.

**Annotation Preparation.** To ensure consistency across all evaluation tasks, we standardize the evaluation data by converting all samples into a unified multi-modal conversation format and removing potential information leakage. This preprocessing involves: converting numeric class labels to textual descriptions in COCO (Lin et al., 2014); removing explicit numerical references from text descriptions in CountBench

Table 9: Statistics of evaluation benchmarks. We report the number of instances for detection and segmentation tasks. The reported numbers combine validation and test splits where applicable.

| Type | Data | # of samples |
|------|------|--------------|
| Det | COCO | 36,781 |
| | RefCOCO | 5,786 |
| | RefCOCO+ | 5,060 |
| | RefCOCOg | 7,596 |
| Seg | RefCOCO | 1,975 |
| | RefCOCO+ | 1,975 |
| | RefCOCOg | 5,023 |
| | ReasonSeg | 979 |
| Count | Pixmo-Count | 1,064 |
| | CountBench | 504 |
| **SUM** | | 66,023 |

(Paiss et al., 2023); applying consistent formatting across all datasets to maintain evaluation fairness.

**Evaluation Metrics.** For object detection on COCO, we adopt the standard AP metric computed using the COCO API (Team, 2014). For referring object grounding on RefCOCO(+/g), we use bbox AP, which measures detection accuracy at an IoU threshold of 0.5. For object segmentation on RefCOCO(+/g) and ReasonSeg, we use gIoU, computed as the mean IoU across all segmentation masks. For counting tasks, we use count accuracy as evaluation metric.

**Statistics and Visualization.** We show the statistic data in Table 9. For detection and segmentation tasks, we report the number of valid instances. For counting tasks, we provide the total number of test samples. Our evaluation comprises a total of 66,023 test samples, covering three fundamental visual perception task types and 10 specific tasks. We visualize some examples in Figure 8.

# C  MORE TRAINING DETAILS

The training is conducted on a single node with 8 GPUs and the entire training process takes 6 hours. The peak GPU memory usage is approximately 80 GB, though this can be adjusted through hyperparameters such as memory_utilization in VeRL (Sheng et al., 2024). The reward converges at around 100 steps, and the best checkpoint is typically obtained at around 200 steps.

## D    EXPERISSION LEVEL EVALUATION ON REFCOCO(+/G)

Our primary evaluation, detailed in Appendix B, reports instance-level performance. However, since the RefCOCO(+/g) benchmarks provide multiple expressions per image, we additionally present expression-level results in Table 10.

Table 10: Performance comparison on expression-level RefCOCO(+/g) tasks. Results with * are cited from the Qwen2.5-VL report but are not reproducible in our environment.

| Method | Detection | | | | | | SUM |
| | RefCOCO | | RefCOCO+ | | RefCOCOg | | |
| | val | testA | val | testA | val | test | |
|---|---|---|---|---|---|---|---|
| Qwen2.5-VL-7B* | 90.0 | 92.5 | 84.2 | 89.1 | 87.2 | 87.2 | 530.2 |
| Qwen2.5-VL-7B | 89.0 | 92.0 | 83.2 | 88.3 | 86.4 | 86.5 | 525.4 |
| VisionReasoner-7B | 89.1 | 91.0 | 85.0 | 87.6 | 87.6 | 88.5 | **528.8** |

## E    TASK ROUTER

In order to identify users' instruction automatically during inference, we also train a TaskRouter. The TaskRouter $\mathcal{F}_{\text{router}}$ is a pure language model that processes textual instructions. For any given instruction $\mathbf{T}$, TaskRouter performs a semantic analysis and outputs a task classification $\mathbf{C}$ into one of four predefined fundamental task categories. This mapping can be formally expressed as:

$$\mathbf{C} = \mathcal{F}_{\text{router}}(\mathbf{T}). \tag{5}$$

We train TaskRouter using the GRPO algorithm (Shao et al., 2024), providing reward signals exclusively upon correct task classification. We evaluate the effective of the TaskRouter and results are shown on Table 11. The task classification dataset is constructed from diverse visual perception datasets and AI-generated samples. For datasets that include textual instructions (*e.g.*, RefCOCOg), we retain their original instructions and corresponding fundamental task categories. Additionally, for each fundamental task type, we employ ChatGPT (OpenAI, 2023) to generate instructions and target categories. The

Table 11: Comparison on the task classification.

| Model | Accuracy |
|---|---|
| Qwen2.5-1.5B | 46.3 |
| TaskRouter-1.5B | **99.1** |

final dataset comprises 20,000 training samples and 4,000 test samples. Although the state-of-the-art Qwen2.5 (Yang et al., 2024) demonstrates strong performance in instruction following and zero-shot task classification, its accuracy drops below 50% in our complex scenario. In contrast, our task router module, trained using reinforcement learning, achieves significantly better performance.

## F    USER PROMPT TEMPLATE

To guide the policy model toward generating desired outputs during exploration, we employ the user prompt template presented in Table 12. This prompt template is inspired by DeepSeek-R1-Zero (Guo et al., 2025) and Seg-Zero (Liu et al., 2025a).

## G    QUALITATIVE RESULTS

We visualize results on Figure 9. Our model generates comprehensive reasoning processes for all tasks while producing expected outputs. We find that VisionReasoner can effectively distinguish between similar objects, as shown in the visual grounding and referring expression segmentation. VisionReasoner also accurately localize multiple targets, as shown in object detection and counting. We also observe that the length of the reasoning process adapts dynamically: more intricate image-query pairs elicit detailed rationales, while simpler inputs result in concise explanations.

Table 12: User Prompt. "*{Question}*" is replaced by user questions during training and inference.

---

**User Prompt**

"Please find "*{Question}*" with bboxs and points."
"Compare the difference between object(s) and find the most closely matched object(s)."
"Output the thinking process in <think> </think> and final answer in <answer> </answer> tags."
"Output the bbox(es) and point(s) inside the interested object(s) in JSON format."

```
i.e. <think> thinking process here </think>
<answer>[{"bbox_2d": [10,100,200,210], "point_2d" [30,110]},
{"bbox_2d": [225,296,706,786], "point_2d": [302,410]}]</answer>
```

---

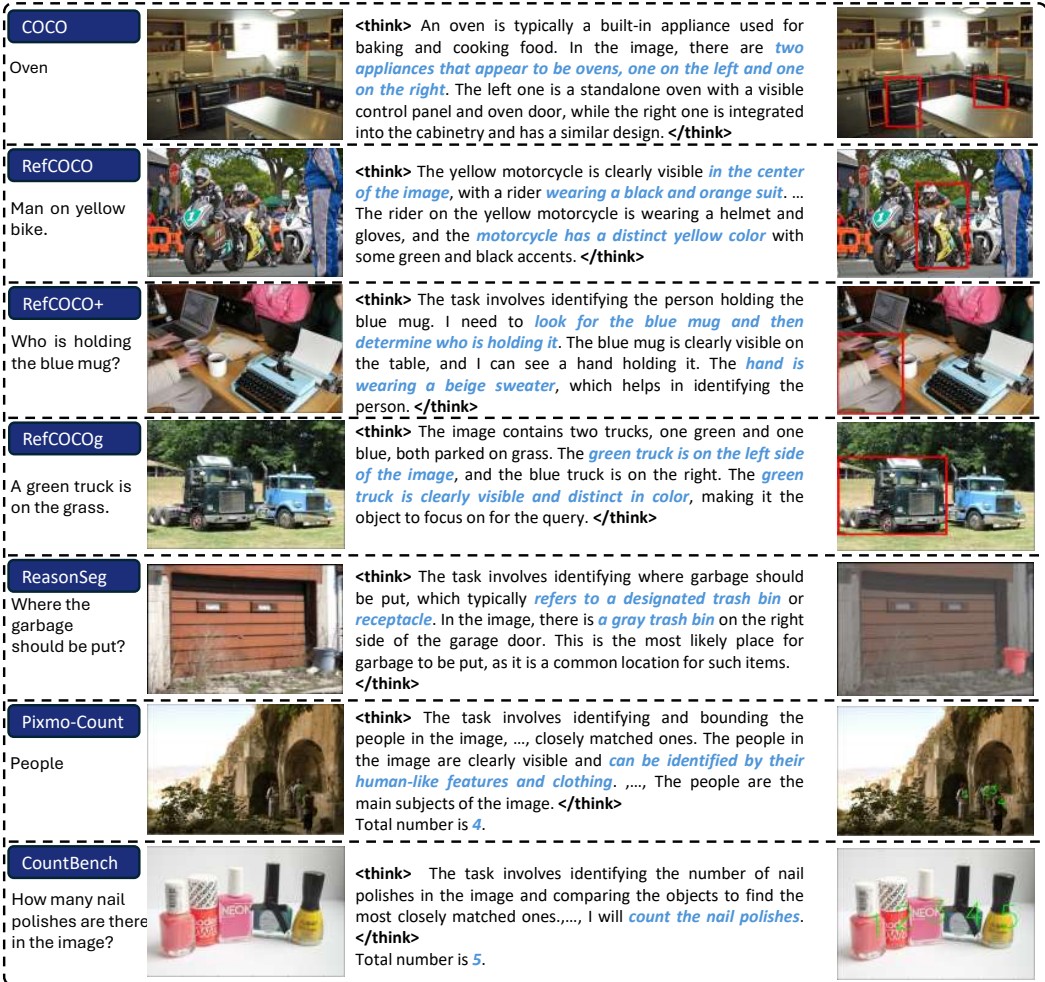

Figure 9: Qualitative results on different tasks. Zoom in for better visualization.

## H EXTENSION

**More Applications.** Our VisionReasoner can be easily extended to other visual perception tasks that need reasoning. We just need to add a light-weight module for different output format. The intermediate output $\{\mathbf{B}_i\}_{i=1}^{N}$ and $\{\mathbf{P}_i\}_{i=1}^{N}$ serve as bridge to connect other modules. Figure 10 shows our extension to referring expression pose estimation and referring expression depth estimation.

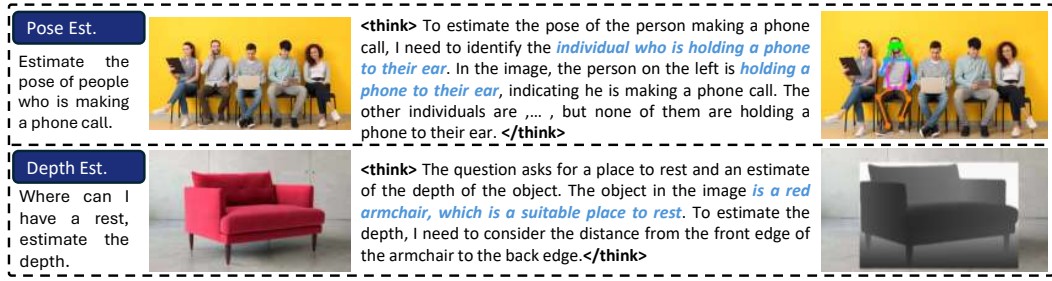

Figure 10: Extending VisionReasoner to more visual perception tasks.

**Hybrid Mode.** We can also employ a hybrid mode, that is directly using traditional visual models (e.g. Yolo-World (Cheng et al., 2024)) for simple categorical instruction (i.e. bird) and VisionReasoner for complex instructions (i.e. 'Where can I have a rest?').

## I  CONCEPTS CLARIFICATION

We formally define the key terms used in this work. As illustrated in Table 13, our hierarchical task formulation adopts the COCO dataset (Lin et al., 2014) as a representative example:

Table 13: Key Terminology and Definitions.

| Concept | Definition |
|---|---|
| Fundamental Task Types | Reformulated task categories (*e.g.*, detection) |
| Task Type | Task category (*e.g.*, object detection) |
| Task | Concrete benchmark (*e.g.*, COCO object detection) |

## J  DETAILS OF TASK REFORMULATION

Within our framework, we categorize task types as illustrated in Table 14 and Table 15. These tables highlight our grouping of task types based on their similarities. It is important to note that although this taxonomy covers a broad range of task types, the current implementation of VisionReasoner is evaluated on only 10 representative tasks, with comprehensive evaluation of all task types reserved for future research.

Table 14: Fundamental Task Types: Counting and Visual Question Answering.

| Counting | VQA |
|---|---|
| Object Counting | Visual Question Answering (VQA) |
| Crowd Counting | Classification |
| Density Estimation | Image Captioning |
| Pedestrian Detection | Question Answering |
| Crowd Estimation in Dense Scenes | Visual Reasoning |
| Traffic Counting in Surveillance | Visual Question Answering |
|  | Relational Reasoning |

Table 15: Fundamental task types: Detection and Segmentation.

| Detection | Segmentation |
|---|---|
| Visual Grounding | Semantic Segmentation |
| Object Detection | Instance Segmentation |
| 2D Object Detection | Lane Detection |
| Small Object Detection | 2D Semantic Segmentation |
| Defect Detection | Medical Image Segmentation |
| Face Detection | Human Part Segmentation |
| License Plate Detection | Action Segmentation |
| Anomaly Detection | Video Object Segmentation |
| Human Detection | Referring Expression Segmentation |
| Surgical Tool Detection | Saliency Detection |
| Dense Object Detection | Salient Object Detection |
| Open World Object Detection | Semantic Segmentation of Remote Sensing Imagery |
| Zero-Shot Object Detection | Crack Segmentation |
| Animal Action Recognition | Action Unit Detection |
| Robotic Grasping | RGB Salient Object Detection |
| Object Localization | Boundary Detection |
| Hand Detection | Crack Segmentation for Infrastructure |
| Visual Relationship Detection | Surgical Tool Segmentation |
| Open Vocabulary Object Detection | |
| Oriented Object Detection | |
| Object Detection in Indoor Scenes | |
| Object Detection in Aerial Images | |
| Person Search | |
| Object Recognition | |

