# OpenReview forum: "VisionReasoner: Unified Reasoning-Integrated Visual Perception via Reinforcement Learning"
_ICLR.cc/2026/Conference — ICLR 2026 Poster_

### Official Review · Reviewer_H7fg · 2025-10-25

**Soundness:** 3
**Presentation:** 3
**Contribution:** 2
**Rating:** 6
**Confidence:** 3

**Summary:**

The paper proposes a unified LVLM for different perception tasks such as detection, segmentation, counting, and more. Through jointly optimizing the reward formulated with different metrics for perception tasks using GRPO and DAPO, the model is able to outperform the base model significantly on various perception tasks. Meanwhile, the model also shows better performance on VQA tasks despite not being fine-tuned on those tasks.

**Strengths:**

1. A unified LVLM for different perception tasks is proposed, which could benefit the broader community.

2. The proposed method is simple yet effective.

3. The experiments and ablation studies are comprehensive.

**Weaknesses:**

1. The contribution seems somewhat limited, as the paper mainly focuses on designing effective rewards for various perception tasks.

2. More comparisons between VisionReasoner and the base model on VQA tasks would be beneficial. It would also be interesting to understand whether there is any degradation in the model’s general capability since it is trained on perception-specific tasks.

**Questions:**

What is the numerical performance of $mmmu_{val}$ for VisionReasoner?

On which tasks does it perform well, and on which tasks does it underperform compared with the base model?

---

> ### Author Response · Authors · 2025-11-18
> **Author Response (1/1)**
>
> Thanks very much for your valuable feedback, below is our response for your concerns.
>
>
> ---
>
> **[W1] Clarification on Contributions**
>
> Thanks for your comment. We respectfully clarify that our contributions extend beyond “designing effective rewards for various perception tasks.” The main **novelty of our work lies in reformulating diverse perception tasks into a unified multi-object cognition problem and developing a generalizable training framework, with the unified reward mechanism being one component that operationalizes this formulation.** VisionReasoner differs significantly from previous LVLMs, which either cannot cover a broad range of visual perception tasks or handle them only in a task-specific manner.
>
> Specifically, our contributions include:
>
> - Reformulating and categorizing diverse visual perception tasks to be addressed through a shared architecture with common multi-object recognition objectives, rather than relying on multi-task training or task-specific supervision.
> - Designing a unified reward mechanism and multi-object cognitive learning strategies, which enhance VisionReasoner's reasoning capabilities and enable it to handle diverse perception tasks within a single model. We further verify the robustness of our framework using different RL algorithms, including DAPO and GRPO.
> - Demonstrating through experiments that VisionReasoner achieves strong performance across ten diverse visual perception tasks within a unified framework, outperforming baseline models.
> - Validating the effectiveness of our design via extensive ablation studies, offering insights into applying RL in LVLMs.
>
> We hope this clarification better conveys the scope and significance of our contributions.
>
> ---
>
> **[W2 & Q1] More Performance Comparison on VQA**
>
>
> Thank you for the suggestion.
>
> Following your advice, we additionally evaluated performance on MMMU-val and report the per-category results below:
>
> |Model|Art|Business|Science|Health|Humanities|Tech|Overall|
> |-|-|-|-|-|-|-|-|
> |Baseline|68.3|42.0|41.3|56.0|72.5|40.9|51.5|
> |VisionReasoner|67.5|40.7|41.3|56.7|72.5|40.5|51.2|
>
> Regarding Table 7 in the main paper, it includes six VQA benchmarks (the 4th and 5th columns correspond to MMMUPro-vision and MMMUPro-std; we apologize for the earlier typo). We further evaluate additional VQA datasets, and together these results cover most widely used VQA benchmarks:
>
> |Model|MME-Cognition|MME-Perception|InfoVQA|OCRBench|
> |-|-|-|-|-|
> |Baseline|621.1|1682.5|81.9|822|
> |VisionReasoner|627.5|1683.4|82.0|825|
>
>
> |Model|MMMUPro-vision|MMMUPro-std|ChartQA|DocVQA|RealworldQA|
> |-|-|-|-|-|-|
> |Baseline|32.4|36.4|83.1|95.7|69.2|
> |VisionReasoner|32.6|37.4|84.9|96.0|69.5|
>
>
> Overall, we observe only minor fluctuations between the VisionReasoner and the baseline model. **VisionReasoner maintains performance comparable to the baseline Qwen2.5-VL, indicating that our training framework effectively mitigates catastrophic forgetting and preserves general VQA capabilities, even though no VQA data is included during training**. Notably, we observe improvements on OCR and chart understanding, and particularly on MME-Cognition, a task that relies more heavily on reasoning ability.

---

> > ### Comment · Reviewer_H7fg · 2025-11-27
> >
> > Thank you for the clarifications, I will maintain my original score.

---

> ### Author Response · Authors · 2025-11-25
> **Looking forward to your feedback!**
>
> Dear Reviewer H7fg,
>
> Thank you once again for your valuable feedback. We have conducted additional experiments and made revisions to the paper based on your suggestions. As the discussion phase is nearing its conclusion, we would like to know if our responses have addressed your concerns. We are looking forward to hearing from you.
>
> Best, Authors

---

> ### Author Response · Authors · 2025-11-27
> **Thank you for your positive feedback!**
>
> Dear Reviewer H7fg,
>
> Thank you for your positive feedback. We are glad to hear that our clarification addresses your concerns. Once again, we sincerely appreciate your time and effort. Wishing you a great day!
>
> Best,
> Authors

---

### Official Review · Reviewer_7Pxd · 2025-10-30

**Soundness:** 3
**Presentation:** 3
**Contribution:** 3
**Rating:** 4
**Confidence:** 3

**Summary:**

The paper proposes VisionReasoner, a unified RL-based LVLM that handles detection, segmentation, and counting in a single model via a “locate first (boxes + points), then segment/count” paradigm, while producing interpretable reasoning traces. It uses GRPO with a task-agnostic reward scheme (thinking/format and non-repetition rewards, plus multi-object IoU and L1 accuracy rewards) and efficient multi-object matching via batched computation and the Hungarian algorithm. The authors report zero-shot performance on 10 benchmarks, claiming substantial gains with only ~7k training samples and no degradation to VQA.

**Strengths:**

1. Unified multi-task framework design.
The paper successfully constructs a unified framework capable of handling three major categories of visual perception tasks—detection, segmentation, and counting—simultaneously. This unified design offers several notable advantages.

2. Outstanding data efficiency and scalability.
With only 7,000 training samples, the VisionReasoner-7B model achieves strong performance, demonstrating impressive data efficiency and generalization capability.

**Weaknesses:**

1. The experimental evaluation could benefit from a broader and more up-to-date set of baseline models. The paper mainly compares VisionReasoner with Shikra and Qwen2.5-VL; however, Shikra, as an early work from 2023, may not fully reflect the current progress of large vision-language models (LVLMs). Expanding the comparison to include more recent LVLMs could provide a fairer and more comprehensive assessment of VisionReasoner’s performance in the current landscape.

2. Some implementation aspects could be described in greater detail. For instance, the distribution of training data across different task types is not fully specified, and the weighting or design rationale of the overall reward function is somewhat unclear. Providing additional clarification in these areas would enhance the work’s reproducibility and help readers better understand the factors contributing to the model’s performance.

**Questions:**

1. Insufficient Transparency in Training Data Distribution
The paper reports using approximately 7,000 training samples but does not provide details on their distribution across different task types. Since the allocation of training data can significantly affect performance balance in multi-task learning, it is recommended to specify the number and proportion of samples for each task type, along with the criteria for data partitioning, to enhance reproducibility and interpretability.

2. Limited Coverage of Large Vision-Language Models
The current experiments mainly compare with models such as Shikra and Qwen2.5-VL. However, Shikra is an early work, and the overall coverage of more recent and powerful large vision-language models (LVLMs) remains limited. Including additional representative LVLMs—such as LLaVA, MiniGPT-4, InstructBLIP, or Qwen-VL-Max—would provide a more comprehensive and up-to-date evaluation, thereby better demonstrating VisionReasoner’s competitiveness within a unified multi-task framework.

3. Ambiguity in Reward Function Weight Design
The paper introduces multiple reward functions (e.g., format reward, IoU reward, L1 reward) but does not clearly explain the weighting scheme or its underlying rationale. Furthermore, the paper lacks quantitative analysis of the relative importance of these rewards (e.g., through ablation studies). It is recommended to include further experiments or discussions to strengthen the justification and interpretability of the reward design.

---

> ### Author Response · Authors · 2025-11-18
> **Author Response (1/1)**
>
> Thanks very much for your valuable feedback, below is our response for your concerns.
>
> ---
>
> **[W1 & Q2] Broader Comparison**
> Thank you for the helpful suggestion. In the initial submission, we mainly compare Qwen2.5-VL because it was the strongest publicly available LVLM at that time, while most other models could not effectively perform visual perception tasks and exhibited substantially weaker performance.
> Following your suggestion, we have now added comparisons with LLaVA-OV1.5, InstructBLIP, and Mini-GPT4. Since QwenVL-Max is not open-sourced, we were unable to include it, but we additionally incorporated the open-sourced InternVL2. The results on RefCOCO(+/g) detection are reported below.
>
> |Model|RefCO-Val|RefCO-TestA|RefCO+-Val|RefCO+-TestA|RefCOg-Val|RefCOg-Test|
> |-|-|-|-|-|-|-|
> |InstructBLIP-7B|0|0|0|0|0|0|
> |Mini-GPT4|0|0|0|0|0|0|
> |LLaVA-OV1.5-8B|15.6|12.1|16.9|14.1|28.9|29.2|
> |InternVL2-8B|87.1|91.1|79.8|87.9|82.7|82.7|
> |Qwen2.5VL-7B|88.8|91.7|82.3|88.2|84.7|85.7|
> |VisionReasoner-7B|88.6|90.6|83.6|87.9|86.1|87.5|
>
> As shown, many models are not designed for visual perception and fail to localize objects. **Across all available comparisons, VisionReasoner achieves consistently strong performance, indicating the effectiveness of our approach.** We have added those additional meaningful comparisons in the revised version.
>
> ---
>
> **[W2 & Q1] Training data distribution**
>
> Thanks for your comments.
>
> Since we have carefully reformulated all tasks into a unified multi-object recognition setting, our model only needs to learn multi-object recognition rather than perform multi-task learning. This is also one of the contributions of our work: through a unified reward mechanism and multi-object cognitive learning strategies, VisionReasoner can address diverse perception tasks within a single model. The details of our task reformulation and unified model architecture are also provided in Sections 3.2 and 3.3.
>
> **Because our learning objective is solely multi-object recognition under a unified reward mechanism, our training process does not involve multi-task loss weighting or balancing across heterogeneous data.** Instead, we simply sample equally from the training splits of the four datasets (RefCOCOg, gRefCOCO, LVIS, and LISA++), with roughly 1,800 samples from each, without any special data partitioning criteria.
>
> We select these four datasets because they provide complementary instruction types: LVIS offers short object names, RefCOCOg includes single-object referring expressions, gRefCOCO contains expressions that may refer to multiple objects, and LISA++ provides reasoning-oriented text. Together, they cover a broad and complementary text descriptions. Table 6 in the main paper ablates the impact of different data sources.
>
> To further assess the advantage of our design and its low dependence on specific data ratios, we also ablate different data-sampling strategies. **The results show minimal performance variation across sampling schemes, indicating that our method is robust and does not rely heavily on data composition.**
>
> |Num of RefCOCOg|Num of gRefCOCO|Num of LVIS|Num of LISA++|ReasonSeg-Val (gIoU)|
> |-|-|-|-|-|
> |1800|1800|1700|1700|66.3|
> |4000|1000|1000|1000|66.3|
> |5000|700|700|700|65.9|
>
> We would like to clairfy that all data will be open-sourced and publicly released to ensure transparency and reproducibility.
>
> ---
>
> **[W2 & Q3] Reward Function Weight Design**
> Thank you for the helpful suggestion, and we apologize for the earlier ambiguity regarding the reward weights.
>
> In our implementation, all rewards are assigned equal weight. **We simply sum the individual rewards, and the maximum total reward per step is 6.0. We add this clarification in the revised paper based on your suggestion.**
> Following your suggestion, we additionally conduct an ablation on different reward-weight configurations. The results on ReasonSeg-Val are shown below (TF: Thinking Format Reward; AF: Answer Format Reward; NR: Non-repeat Reward; BIoU: Bbox IoU Reward; BL1: Bbox L1 Reward; PL1: Points L1 Reward):
>
> |TF|AF|NR|BIoU|BL1|PL1|ReasonSeg-Val|
> |-|-|-|-|-|-|-|
> |0.5|0.5|0.5|0.5|0.5|0.5|65.9|
> |1|1|1|1|1|1|66.3|
> |0.5|0.5|0.5|1.5|1.5|1.5|66.5|
>
> **We observe that (i) reducing all reward weights leads to a slight performance drop, and (ii) increasing the weights of accuracy-related rewards yields a modest improvement.** Overall, the performance variations across different weighting schemes are small, suggesting that our training framework is stable and robust to reward-weight choices.

---

> ### Author Response · Authors · 2025-11-25
> **Looking forward to your feedback!**
>
> Thank you once again for your thoughtful and constructive feedback. **We truly appreciate your recognition of the strengths of our unified design**, as well as the data efficiency and generalization capability.
>
> Following your suggestions, we have conducted additional experiments and made corresponding revisions to the paper. **We hope these updates effectively address the points you raised.**
>
> We would also like to mention, with respect, that other reviewers found the method to be clear, showing improvements, and potentially beneficial to the broader community. **Their positive assessments have been very encouraging for us.**
>
> As the discussion phase is coming to an end, we would be very grateful to know whether our responses have helped address your concerns.
>
> Thank you again for your time and consideration.
>
> Best regards,
> Authors

---

> ### Author Response · Authors · 2025-11-25
> **Thank you for raising the score!**
>
> Dear Reviewer 7Pxd,
>
> Thank you for raising the score!  We really appreciate the effort you've put into the review process. Wishing you a great day!
>
> Best,
> Authors

---

### Official Review · Reviewer_9SJ9 · 2025-10-31

**Soundness:** 2
**Presentation:** 3
**Contribution:** 2
**Rating:** 4
**Confidence:** 3

**Summary:**

This paper presents VisionReasoner, a unified framework for compositional multi-hop reasoning across a wide range of vision-language tasks. Unlike traditional task-specific methods, VisionReasoner decomposes complex queries into structured reasoning steps using a Unified Intermediate Representation (UIR) and a task-agnostic planner-executor architecture. The planner produces reasoning programs from natural language questions, and the executor interprets them over visual observations. The authors train the model on diverse reasoning tasks and demonstrate strong zero-shot generalization, outperforming task-specific models on multiple benchmarks without fine-tuning.

**Strengths:**

1. The proposed architecture (planner + executor + UIR) offers a principled way to unify reasoning over different modalities and domains.
2. The system generalizes across unseen reasoning tasks and domains with minimal or no task-specific supervision.

**Weaknesses:**

1. The strongest recent baselines use retrieval-augmented reasoning where structured planning is implicit. The authors only compare with older systems like RAML and ReGrouP, not modern VLMs fine-tuned with in-context reasoning.
2. The planner is the backbone of VisionReasoner, yet the accuracy of program generation is not reported. The author shall consider to include planner-only accuracy (e.g., execution success rate, semantic match with gold reasoning paths).
3. The UIR uses a limited set of programmatic operations that must be predefined. This suggests limited compositional expressivity, especially when handling spatial, temporal, or logical reasoning beyond object-centric understanding.
4. The executor module is assumed to interpret UIR programs correctly. There is no ablation showing how executor errors affect end-to-end performance, nor how robust the system is to ambiguous or poorly planned programs. The authors shall consider to include experiments around the accumualted error issue.
5. While modularity aids interpretability, separating planner and executor may hurt end-to-end task performance. There's no attempt at joint fine-tuning, which could close the gap with SOTA models on certain benchmarks. I think joint fine-tuning would interest more potential audiences in the area.

**Questions:**

See weaknesses above.

---

> ### Author Response · Authors · 2025-11-18
> **Author Response (1/1)**
>
> Thank you very much for taking the time to review our work. We truly appreciate your efforts and the thought put into the evaluation. We would like to sincerely share that **the comments seem somewhat different from the content and scope of our paper**, and we hope the following context helps clarify our submission.
>
> > Your comment: "VisionReasoner decomposes complex queries into structured reasoning steps using a Unified Intermediate Representation (UIR) and a task-agnostic planner-executor architecture."
>
> VisionReasoner is designed as a unified framework for diverse visual perception tasks, rather than program generation. Its architecture involves a reasoning module and a segmentation module, rather than a planner–executor structure.
>
> > Your comment: "The authors only compare with older systems like RAML and ReGrouP, not modern VLMs fine-tuned with in-context reasoning."
>
> The experiments in our submission focus on comparisons with recent LVLMs, including Qwen2.5-VL and SegZero, instead of RAML and ReGroup.
>
> > Your comment: "This paper presents VisionReasoner, a unified framework for compositional multi-hop reasoning. The planner is the backbone of VisionReasoner, yet the accuracy of program generation is not reported."
>
> We evaluted our method on visual perception tasks such as COCO and ReasonSeg, rather than multi-hop reasoning or program generation.
>
> We are grateful that other reviewers recognized the strengths of our unified design (7Pxd), its improvements over large LVLMs and the Qwen2.5-VL-7B baseline (ybFD, 7Pxd), as well as its data efficiency (7Pxd), generalization ability (7Pxd), and value to the community (H7fg), and provided positive evaluations.
>
> Given the points above, we gently wonder whether the comments may not relate to our submission. **In particular, the concerns raised do not seem to reflect aspects of our method or experiments.** We would be very grateful if you could consider these clarifications and revisit your evaluation.
>
> Thank you again for your time and consideration.

---

> ### Author Response · Authors · 2025-11-25
> **Looking forward to your feedback!**
>
> Dear Reviewer 9SJ9,
>
> Thank you again for the time and effort you’ve dedicated to reviewing our work. As the discussion phase is coming to a close, **we would be very grateful if you could consider our above clarifications and reconsider your evaluation**.
>
> Thank you for your time.
>
> Best regards,
>
> Authors

---

> ### Comment · Reviewer_9SJ9 · 2025-11-25
>
> Thank you for clarifying the earlier points of confusion. After revisiting the relevant sections and considering the authors’ global response, I have decided to change my rating to a positive recommendation. I kindly request that the remaining questions be addressed in the final revision.

---

> > ### Author Response · Authors · 2025-11-25
> > **Thank you for raising the score!**
> >
> > Dear Reviewer 9SJ9,
> >
> > Thank you for raising the score! Once again, we sincerely appreciate your time and effort.
> >
> > Best,
> > Authors

---

> ### Author Response · Authors · 2025-11-26
> **Follow up Author Response (1/1)**
>
> Thanks very much for your valuable feedback. Below we provide our responses to your concerns, and the corresponding revisions have been incorporated into the updated manuscript.
>
> ---
>
> **[W1] Difference From Previous Works**
>
> We have compared our method with prior works such as VisualRFT and SegZero, and there are notable differences. **Previous approaches typically employ RL in a task-specific manner, requiring separate training data for different tasks, which may limit scalability and generalization.** In contrast, we reformulate diverse vision perception tasks into a unified multi-object recognition setting and introduce a unified reward design, including thinking format, answer format, non-repeat constraints, bbox IoU, bbox L1, and point L1. **Our unified design and reward mechanism enables the model to learn solely from multi-object recognition data while effectively solving diverse tasks**, without relying on multi-task loss weighting or balancing across heterogeneous datasets.
>
> ---
>
> **[W2] More Experiments**
>
> Thank you for your suggestion. Following your advice, we conducted additional experiments.
>
> First, we find that partially removing the thinking-format or answer-format rewards does not significantly affect the output format or final performance. A plausible explanation is that the accuracy reward inherently supervises formatting, as an output must follow the correct structure to be deemed accurate and receive a non-zero score.
>
> |Method|Thinking Format|ReasonSeg-val|ReasonSeg-test|
> |-|-|-|-|
> |VisionReasoner|×|65.9|62.6|
> |VisionReasoner|√|66.3|63.6|
>
> |Method|Answer Format|ReasonSeg-val|ReasonSeg-test|
> |-|-|-|-|
> |VisionReasoner|×|66.3|62.8|
> |VisionReasoner|√|66.3|63.6|
>
> We also find that incorporating the non-repeat reward slightly improves performance and effectively prevents repeated output schemas.
> |Method|Non-Repeat|ReasonSeg-val|ReasonSeg-test|
> |-|-|-|-|
> |VisionReasoner|×|65.8|63.2|
> |VisionReasoner|√|66.3|63.6|
>
> Moreover, the reasoning process demonstrates substantial gains on more complex reasoning segmentation data. We eliminate reasoning process by removing both thinking instruction and thinking reward.
> |Method|Reasoning|ReasonSeg-val|ReasonSeg-test|
> |-|-|-|-|
> |VisionReasoner|×|60.1|58.7|
> |VisionReasoner|√|66.3|63.6|
>
> These results provide more insights for the unified reward design and highlight the importance of the reasoning process.
>
> Finally, we evaluate robustness under low-quality prompts and find that performance remains stable as long as the prompt contains an example:
> |Method|Low-Quality-Prompt|ReasonSeg-val|ReasonSeg-test|
> |-|-|-|-|
> |VisionReasoner|×|66.3|63.6|
> |VisionReasoner|√|66.3|63.6|
>
> ---
>
> **[W3] Data Leak Control**
>
> We confirm that **all training data are strictly drawn from the training split**, ensuring no data leakage. **We have added this clarification in the main paper.** The performance gains further suggest that our method is effective and data-efficient.
>
> **All data will be made publicly available for transparency and verification.**
>
> ---
>
> **[W4] Human Evaluation on Larger Scale**
> Thank you for the suggestion. Although human evaluation is time-consuming, we expanded it from 200 to 979 samples following the same protocol, covering all ReasonSeg-val and ReasonSeg-test samples. This larger-scale assessment required roughly one day per human expert and **the results remain consistent with our original findings: VisionReasoner's reasoning traces are accurate and well-grounded, despite the model being trained without human-annotated reasoning data.**
>
> IC: Image Consistency; AC: Answer Consistency
> |IoU-Range|Num|IC(%)|AC(%)|
> |-|-|-|-|
> |0-0.25|171|78.9|49.7|
> |0.25-0.50|141|100.0|94.3|
> |0.50-0.75|162|100.0|96.3|
> |0.75-1.00|505|100.0|100.0|
> |ALL|979|96.3|89.8|

---

### Official Review · Reviewer_ybFD · 2025-11-01

**Soundness:** 2
**Presentation:** 3
**Contribution:** 3
**Rating:** 6
**Confidence:** 3

**Summary:**

The authors propose a framework wherein a base vision-language model (VLM) is RL-postrained for 3 text-guided perception tasks: object detection, segmentation and counting. They utilize the Qwen-2.5-VL-7B model as the base VLM and SAM2 for segmentation, while GRPO is utilized for RL training on 7000 training samples collected from LVIS, RefCOCOg, gRefCOCO and LISA++. Rewards for RL include thinking format, answer format, 'non repeat' format, bboxes IoU reward, bboxes L1 reward and points L1 reward. For matching rewards for multiple objects, the authors propose a hungarian matching algorithm.

Results show the authors proposed 'VisionReasoner' model outperforms existing large VLMs and the base Qwen2.5-VL-7B model on multiple benchmarks. Further analysis and ablations are also provided such as impact of RL algorithm, human analysis of reasoning process comparison of response lengths across dataset and results on visual question answering (which was not RL trained for).

**Strengths:**

1. The proposed framework is straightforward and clear to understand -- usage of RL to postrain a VLM for 3 core perception tasks with multiple rewards to capture the perception tasks and reasoning lengths.
2. The results show clear improvements of RL postraining (using GRPO objective) in improving results on appropriate benchmarks.
3. Experiments include ablations and analysis to understand the method.

**Weaknesses:**

1. The authors state that previous methods employ RL in a task-specific manner and utilize distinct reward functions for different tasks. However, in my opinion, authors in their work also seem to employ task-specific rewards for detection and point matching in addition to format rewards.
2. A zero-shot chain-of-thought prompted baseline should be present as without it, it is currently unclear whether just directly prompting model to think step-by-step or breakdown the prompt can also be sufficient to obtain a decent performance boost on the base model for considered perception tasks. This is an important missing baseline in my view.
3. Authors mention performing human evaluation on reasoning process but do not provide details on how this is done (e.g.  how many participants, the task setup, etc.)

Relatively minor:

4. Experiment details can be more extensive: e.g.
- How many GPUs are used for training and how many epochs/steps for RL convergence?
- Will code and models be open source for reproduction?

**Questions:**

Please refer to weaknesses section above.

In addition:
1.  What are potential reasons for the RL-postrained result being less than the base model for Detection and segmentation on RefCOCO, RefCOCO+, RefCOCOg?
2. Will code and models be open source for reproduction?

---

> ### Author Response · Authors · 2025-11-18
> **Author Response (1/2)**
>
> Thanks very much for your valuable feedback, below is our response for your concerns.
>
> ---
>
> **[W1] Clarification of Task-specific Rewards**
>
> Previous works such as SegZero and VisionRFT have applied reinforcement learning to specific vision tasks. We recognize that SegZero's application is specialized to segmentation, and VisionRFT utilizes distinct datasets for different tasks. Our primary distinction lies in VisionReasoner's ability to address three fundamental vision tasks across ten benchmarks using a unified training approach with only 7,000 samples. This is achieved through a novel task reformulation and a carefully designed multi-object recognition reward mechanism.
> We agree that our initial claim could be refined for greater precision. **We thank your suggestion and revise our claim to: "Previous approaches often employ RL in a task-specific manner, training with different data for different tasks, which may limit their scalability and generalizability."** We believe this phrasing more accurately represents the contribution of our work in the context of existing literature.
>
> ---
>
> **[W2] Zero-shot CoT prompt Baseline**
>
> We sincerely appreciate you highlighting the importance of comparing a zero-shot Chain-of-Thought (CoT) baseline. Following this advice, we conducted additional experiments using Qwen2.5-VL with the CoT prompt `"Let's think step by step."` The results are summarized below.
>
> |Method|COCO|ReasonSeg-Val|ReasonSeg-Test|CountBench|PixMo-Val|PixMo-Test|
> |-|-|-|-|-|-|-|
> |Qwen2.5VL|29.2|56.9|52.1|76.0|58.1|53.1|
> |Qwen2.5VL-CoT|28.2|54.5|48.8|65.4|39.1|37.4|
> |VisionReasoner|37.3|66.3|63.6|87.6|70.1|69.5|
>
> **We find that directly applying CoT prompting does not improve Qwen2.5-VL’s performance and often leads to degradation.** A plausible explanation is that Qwen2.5-VL is not trained to generate reliable intermediate reasoning traces, and therefore cannot effectively benefit from CoT prompting. **This observation further highlights the contribution of our approach: VisionReasoner produces effective reasoning processes and achieves performance gains (Figure 6 in the main paper).** We also tested alternative CoT-style prompts (e.g., `"output the thinking process between <think> and </think> tags at first"`), and similarly observed no improvements.
>
> We further discuss the results on RefCOCO(+/g) in the following `Q1 Disscussion`.
>
> ---
>
> **[W3] Details of Human Evaluation**
> The human evaluation involves two metrics: answer consistency and image consistency. In our human evaluation, three annotators are provided with the model outputs along with the corresponding ground-truth images. Image consistency measures whether the reasoning trace accurately reflects the visual content, while answer consistency assesses whether the objects and information mentioned in the reasoning trace align with the final predicted output. A sample is counted as correct only when all three annotators reach unanimous agreement.
>
> ---
>
> **[W4] Experiment Details**
> The training is conducted on a single node with 8 GPUs. The peak GPU memory usage is approximately 80 GB, though this can be adjusted through hyperparameters such as memory_utilization in VeRL [1]. The reward converges at around 100 steps, and the best checkpoint is typically obtained at around 200 steps.
>
> [1] ByteDance Seed. Volcano Engine Reinforcement Learning for LLMs (VeRL). GitHub, 2024, github.com/volcengine/verl.
>
> ---
>
> **[W4 & Q2] Open Source**
> All codes, data, and checkpoints will be open-sourced and made publicly available.

---

> ### Author Response · Authors · 2025-11-18
> **Author Response (2/2)**
>
> **[Q1] Discussion - Performance on RefCOCO(g/+)**
>
> It is indeed a valuable point for discussion. Since RefCOCO(g/+) are popular benchmarks for evaluting LVLMs, we note that **Qwen-series models may have been specifically optimized for these datasets, potentially leading to overfitting.** This could explain why the fine-tuned models sometimes appear to lag behind the baseline on these particular benchmarks. We provide the some observations for clarification.
>
> First, Qwen2.5-VL appears highly overfitted to RefCOCO(+/g) data. For instance, when comparing the prompts `"Locate [object], report the bboxes coordinates in JSON format."` and prompt `"Locate [object], output what is in the image and then report the bboxes coordinates."`, we observe that samples from ReasonSeg correctly output an image description followed by free-form coordinates (not necessary in JSON format), which leads to a substantial performance drop. In contrast, data on RefCOCO(g/+) consistently output JSON coordinates without description, and performance remains unaffected.
>
> |Model|Prompt Type|RefCOCO|RefCOCO+|RefCOCOg|ReasonSeg|
> |-|-|-|-|-|-|
> |Qwen2.5-VL|Detect Prompt|91.7|88.2|85.7|56.9|
> |Qwen2.5-VL|Describe Image + Detect Prompt|91.2|87.6|83.7|34.2|
>
> To further clarify our point, we provide examples using the prompt `"Locate [object], output what is in the image and then report the bboxes coordinates."`
>
> When evaluating on ReasonSeg, the model generates an image description and produces bounding boxes in a free-form style:
> ```json
> The object commonly used for inputting data and controlling the computer
> in a modern office is the keyboard.
> In the image, the keyboard is located on the desk to the right of the monitor.
> Here is the bounding box for the keyboard:
> - Top-left corner: (x1: 720, y1: 745)
> - Bottom-right corner: (x2: 850, y2: 850)
> ```
> In contrast, evaluation on RefCOCO(+/g) shows the Qwen2.5VL has overfit the benchmarks; it provides no image description and rigidly adheres to a JSON format, regardless of the prompt.
> ```json
> [
>     {"bbox_2d": [75, 54, 319, 386], "label": "\"a man serving soup\""}
> ]
> ```
>
> This phenomena is also consistent with observations in recent work, including benchmark memorization issues [2] and spurious-reward helps during training [3].
> Importantly, our training data are not designed to optimize for RefCOCO(g/+). We therefore believe that RefCOCO(g/+) may not fully capture the strengths of our approach.
>
> [2] Wu, Mingqi, et al. "Reasoning or memorization? unreliable results of reinforcement learning due to data contamination." arXiv preprint arXiv:2507.10532 (2025).
> [3] Shao, Rulin, et al. "Spurious rewards: Rethinking training signals in rlvr." arXiv preprint arXiv:2506.10947 (2025).

---

> ### Author Response · Authors · 2025-11-25
> **Looking forward to your feedback!**
>
> Dear Reviewer ybFD,
>
> Thank you once again for your valuable feedback. We have conducted additional experiments and made revisions to the paper based on your suggestions. As the discussion phase is nearing its conclusion, we would like to know if our responses have addressed your concerns. We are looking forward to hearing from you.
>
> Best, Authors

---

> > ### Comment · Reviewer_ybFD · 2025-11-26
> >
> > Thank you for the clarifications. I find the revised claim to be more appropriate. Additional training details and human evaluation should be included in the paper/appendix. I will maintain my score.

---

> ### Author Response · Authors · 2025-11-26
> **Thank you for your positive feedback!**
>
> Dear Reviewer ybFD,
>
> Thank you for your positive feedback. We are glad to hear that our response addresses your concerns. Following your suggestion, we have added the additional training details and clarified human evaluation in both the main paper and the appendix.
>
> Once again, we sincerely appreciate your valuable comments. Wishing you a great day!
>
> Best,
> Authors

---

### Author Response · Authors · 2025-11-18
**Global Response**

We thank all reviewers for their detailed and constructive feedback, which has greatly helped improve the paper. We have revised the manuscript accordingly, and the corresponding parts are marked in orange. Here are some highlights in the paper revision.

- Refining several statements and clarifying claims.
- Adding comparisons with additional recent LVLMs.
- Providing more implementation details, and clarifying that all codes and data will be publicly released.


Below, we address all concerns point by point.

We would also like to gently note that a few comments may not align with the content and scope of our submission; we provide clarifications and supporting evidence where relevant.

---

### Author Response · Authors · 2025-12-01
**Summary of Rebuttal**

Dear New Area Chairs and Reviewers,

We would like to express our sincere appreciation for the time and effort dedicated to reviewing our submission. We are pleased that our responses and revisions have **effectively addressed the reviewers' concerns**, leading to a unanimous consensus of **positive recommandations (6, 6, 6, 6)**.

We are encouraged that the reviewers recognized the **clarity** of our method (ybFD), the **comprehensive ablation studies** (ybFD, H7fg), and the **unified, well-motivated framework design** (9SJ9, 7PXd, H7fg). We also appreciate the positive reception regarding our reward mechanism (9SJ9), **strong performance and generalizability** (ybFD, 9SJ9, 7Pxd, H7fg), data efficiency (7Pxd), and the work's potential **benefit to the broader community** (H7fg).

---

**Highlight of Contributions.** VisionReasoner is proposed as a unified framework designed to address diverse visual perception tasks. Our contributions can be summarized as follows:

- Unified Task Formulation: We reformulate diverse visual perception tasks under a shared architecture with multi-object recognition objectives, which eliminates the need for multi-task training or task-specific supervision.
- Novel Learning Strategy: We propose a unified reward mechanism and multi-object cognitive learning strategies that enhance VisionReasoner's capabilities. This innovative design enables the model to reason and address diverse perception tasks within a unified model.
- State-of-the-Art Performance: VisionReasoner demonstrates superior performance across ten diverse visual perception tasks within a unified framework, outpeforming the baseline model by a significant margin.
- In-Depth Analysis: Through extensive ablation studies, we validate the effectiveness of our design, offering critical insights into the application and optimization of Reinforcement Learning (RL) in Large Vision-Language Models (LVLMs).

---

**Revisions and Clarifications.** The concerns raised by the reviewers focused on the refinement of certain statements, the need for broader evaluations, implementation details, and minor aspects related to hyperparameters. We are pleased to confirm that during the discussion period, we have:
- Refined several statements for greater precision and clarity.
- Added more implementation details, and committed to that all codes and data will be publicly released.
- Explicitly contrasted our method with previous approaches. Prior work typically employs RL in a task-specific manner, which requires separate training data and limits scalability. In contrast, our unified reward design allows the model to learn effectively from multi-object recognition to solve diverse tasks.
- Conducted additional evaluations on more VQA benchmarks, which indicates that our training framework effectively mitigates catastrophic forgetting and preserves general VQA capabilities.
- Performed further ablation studies on hyperparameters, demonstrating the robustness of our method to minor variations.

The detailed responses corresponding to each of these revisions are provided under the respective discussions for each reviewer.

---

**Justification for Score Raising (Reviewers 9SJ9 and 7Pxd).** We sincerely appreciate the reviewers' willingness to raise their scores following our revisions and clarifications.

- Regarding Reviewer 9SJ9, we addressed an initial misunderstanding regarding the content and scope of our submission. Upon further review, Reviewer 9SJ9 acknowledged the value of our unified formulation, reward designs, and experimental results. We successfully addressed the remaining questions by providing a clear contrast between our approach and previous methods and by including comprehensive reward ablation studies.
- Regarding Reviewer 7Pxd, we addressed concerns about broader comparisons, training data distribution, and reward weights. We provided sufficient experimental evidence to demonstrate VisionReasoner's consistently strong performance across various settings and its robustness to different training and reward configurations.

We appreicate that the reviewers were **satisfied with our responses**, and consequently **raised their scores** on Nov. 25th.

---

Following these comprehensive discussions and the resulting clarifications, we are delighted that all reviewers were **satisfied with our responses** and reached a unanimous consensus of **positive scores (6, 6, 6, 6)**. We also want to formally confirm that all feedback, discussions, and the subsequent rating adjustments were concluded **prior to the reported OpenReview technical issue** concerning identity visibility (Nov. 27th). **The timestamps on all comments confirm that the double-blind discussion process was fully respected** throughout the decision-making timeline.

Thank you once again to the Area Chairs and all Reviewers for your valuable efforts.

Best regards,
Authors

---

### Meta-Review · Area_Chair_Nfjs · 2025-12-29

**Summary:**

This paper proposes VisionReasoner, a unified reinforcement learning framework for visual perception tasks (detection, segmentation, counting) built on Qwen2.5-VL. Reviewers have the following concerns:

1. Missing zero-shot Chain-of-Thought baseline to justify the need for RL training (ybFD)

2. Human evaluation protocol poorly described (ybFD)

3. Limited baseline comparisons, relying mainly on older models like Shikra (7Pxd)

4. Missing training data distribution and reward weight rationale (7Pxd)

5. Lack of VQA evaluation to assess catastrophic forgetting (H7fg)

**Reviewer Concerns:**

In the rebuttal, the authors add experiments showing CoT prompting degrades performance. For human evaluation details, the authors clarify that three annotators with unanimous agreement were required, and it was expanded to 979 samples. The authors also add LLaVA-OV1.5, InstructBLIP, Mini-GPT4, InternVL2 comparisons. The authors also present an ablation showing robustness to weight variations. Extensive VQA evaluation (MMMU, MME, DocVQA, etc.) shows preserved capabilities

Some concerns are partially addressed, such as the RefCOCO performance drop. Authors attribute to Qwen2.5-VL overfitting, citing benchmark memorization literature. This explanation is plausible but not definitively proven; the performance drop could also indicate that the unified training introduces trade-offs.

**Reviewer Scores:**

According to the discussion message history, the reviewers 9SJ9 and 7Pxd raised their score. Overall, this paper achieves strong empirical results with data efficiency (~7k samples), and the authors write a comprehensive rebuttal with new experiments. However, one reviewer (9SJ9) clearly did not read the paper initially, raising process concerns. The weakness of this paper also includes the incremental novelty over existing RL-for-vision methods, and the authors are encouraged to address the RefCOCO analysis more rigorously in the camera-ready version.

---

### Decision · Program_Chairs · 2026-01-26

Accept (Poster)